# Fifty years of recorded hillslope runoff on seasonally-frozen ground: The Swift Current, Saskatchewan, Canada dataset

Anna E. Coles[1,2], Jeffrey J. McDonnell [1,3], Brian G. McConkey[4]

[1] Global Institute for Water Security; School of Environment and Sustainability, University of Saskatchewan, 11 Innovation Boulevard, Saskatoon, SK S7N 3H5, Canada.

[2] Wilfrid Laurier University, 5007 50th Avenue, Yellowknife, NT X1A 2P8, Canada.

[3] School of Geography, Earth and Environmental Sciences, University of Birmingham, Birmingham, B15 2TT, United Kingdom.

[4] Swift Current Research and Development Centre, Agriculture and Agri-Food Canada, PO Box 1030, Swift Current, SK S9H 3X2, Canada.

*Correspondence to*: Anna E. Coles (anna.coles@usask.ca)

**Abstract.** Long records of hillslope runoff and nutrient concentrations are rare—on seasonally-frozen ground they are almost non-existent. The Swift Current hillslopes at the Swift Current Research and Development Centre on the Canadian Prairies provide such a long-term hydrological record. Runoff, runoff nutrient concentration, snowpack depth, density and water equivalent, soil moisture, and soil nutrient concentration were monitored on the three 5 ha hillslopes over a 50-year period (1962 – 2011). Digital elevation data are available for the three hillslopes at a 2 m horizontal resolution, and, for one of the hillslopes (Hillslope 2), at a 0.25 m horizontal resolution. Runoff from the hillslopes was generated episodically during snowmelt and occasional rainfall events. Hillslope runoff was measured with a 0.61 m H-flume. Daily runoff nutrient concentration data are available for nitrate-N (March 1971 – April 2011), ammoniacal-N (February 1996 – April 2011), and phosphate-P (March-April 1971; June 1991 – April 2011). Snowpack data (snowpack depth, density and water equivalent) were determined via manual snow surveys carried out several times each winter, between January and March, between 1965 and 2011. Gravimetric soil moisture content was measured in October and April each year between 1971 and 2011 at five depth intervals (0-15, 15-30, 30-60, 60-90, and 90-120 cm) at nine points on each hillslope. We provide these hillslope data in two publically-available repositories: 1) 1962-2011 data on runoff, runoff nutrients, snowpack, soil moisture, soil nutrients, and crop and tillage practices at https://doi.org/10.23684/hhn5-rz52; and 2) digital elevation data at https://doi.org/10.20383/101.0117 ( Coles et al., 2018b). Complete climate data recorded at a Environment and Climate Change Canada meteorological station located 390 m from the three hillslopes are publically-available at http://climate.weather.gc.ca/.

## 1 Introduction

Long-term datasets that chronicle hydrological processes within the context of climate and land management change are rare. This is especially true for cold regions with seasonally-frozen ground and where remoteness and inclement weather limit continuous measurements. Long-term experimental datasets that do exist in such regions are diminishing quickly (Laudon et al., 2017). For hillslope-scale long-term experiments over seasonally-frozen ground, such records are particularly uncommon. Here we present an unusually-long 50-year record for a set of three monitored agricultural hillslopes in southwest Saskatchewan, Canada that includes runoff, runoff nutrients, snowpack, soil moisture, and soil nutrients for a 50-year period, during which the hillslopes underwent controlled, comparative changes in agricultural practices.

The hillslope-scale dataset is at the spatial scale intermediate between more traditional long-term catchment-scale datasets (e.g. Water Survey of Canada (WSC) Historical Data, the United States Geological Survey (USGS) National Streamflow Information Program, or the UK National River Flow Archive) and point-scale datasets (e.g. TERENO-SOILCan network (Pütz et al., 2016) or the Reynold's Creek soil lysimeter network (Seyfried et al., 2001)). As such, hillslope-scale measurements of precipitation and runoff, together with internal measurements of soil moisture, snowpack, and soil nutrients offer an opportunity to understand mechanistically the links between climate trends in rainfall and snowmelt inputs and runoff outputs. Further, in terms of land use change in dry agroecosystems, they offer an opportunity to understand how tillage, seeding, and crop rotational practices affect the hydrology of agricultural hillslopes, as well as nutrient delivery to downstream water bodies.

The collation of our long-term hillslope database has been motivated largely by the importance of water management and sustainability for dryland agriculture on seasonally-frozen, snowmelt-dominated terrain. This paper reports on an archive of downloadable data from an experimental, agriculturally-managed hillslope site at Swift Current, Saskatchewan. This dataset documents the characteristics more broadly of the semiarid portions of the northern Great Plains of North America. It serves as a resource for long-term analyses and model formulation, calibration, and testing of land management and climate change effects on agricultural hillslopes.

## 2 The Swift Current hillslopes

The three hillslopes (also referred to as 'watersheds' in the dataset) are located at South Farm (50°15'53" N 107°43'53" W), an agricultural research site of Swift Current Research and Development Centre of Agriculture and Agri-food Canada, approximately 5 km southeast of Swift Current, Saskatchewan (Figure 1). The hillslopes adjoin each other and are rectangular in shape, with areas of 4.25 ha (Hillslope 1), 4.66 ha (Hillslope 2), and 4.86 ha (Hillslope 3). The hillslopes are approximately 150 m wide in the east-west direction and 300-320 m long in the north-south direction. They were artificially diked with raised grassed berms constructed in the 1950s by Agriculture and Agri-food Canada.

Hydrological observations at the hillslopes began in 1962 to address the effects of agricultural land management practices on runoff water supply and quality, chemical transport, and soil erodibility.

The hillslopes have undulating topography and slope towards the north-northwest with gradients of 1-4%. Digital elevation models (DEMs), obtained using a Leica Viva GS15 from 17-18 April 2012, are available for all three hillslopes at a 2 m horizontal resolution (Coles et al., 2018b). These DEMs were obtained by mounting the Leica Viva GS15 rover to a side-by-side vehicle at a height of 2 m above ground surface. The vehicle was driven methodically up and down each hillslope, achieving full coverage of each hillslope (Coles et al., 2018b). The GPS was set to automatically collect points every two meters and at instances when the elevation change exceeded 0.15 meters (Coles et al., 2018b). The survey was referenced to a permanent Agriculture and Agri-food Canada benchmark located near to the hillslopes, to give elevations in meters above sea level. Location data for these DEMs are in the projected coordinate system UTM Zone 13N with GRS 1980 datum. Two additional DEMs, obtained using an Optech ILRIS-LR Terrestrial Laser Scanner in summer 2014, are available for Hillslope 2 at a 0.25 m horizontal resolution (Coles et al., 2018b). These two finer DEMs of Hillslope 2 were obtained from surveys on 7 July 2014 and 24 September 2014, before and after a simulated seeding of the hillslope. They therefore capture the surface micro-topography of a tilled hillslope with random soil clod undulations, as well as a seeding-induced ridge-and-furrow topography. These DEMs were obtained by mounting the laser scanner at the top of a 15 ft scaffolding tower located on the east margin of Hillslope 2, and also from the top of a vehicle parked in the northeast corner of the hillslope. The elevation data are relative elevations (not absolute elevations). Location data for these DEMs are in the projected coordinate system UTM Zone 13N with GRS 1984 datum. Additional information on the generation of the 2 m and 0.25 m DEMs can be found in Coles et al. (2018b).

The soils are Orthic Brown Chernozems (Ayres et al., 1985), specifically Swinton silt loam in the upper four-fifths of the hillslopes, where the solum is developed within loess, and Haverhill loam in the lower fifth of the hillslope, where past erosion has resulted in the solum extending into the underlying loamy till (McConkey et al., 1997). The surficial 0-15 cm soil layer consists of 50.4% silt, 31.4% sand, and 18.2% clay, has a bulk density of 1.22 g cm$^{-3}$, and has a saturated hydraulic conductivity of 14.2 mm hr$^{-1}$ (Coles and McDonnell, 2018). Silt content decreases with depth, while clay content increases with depth. Mean soil depth, as measured with a dynamic cone penetrometer, is 2.65 m (Coles and McDonnell, 2018).

The hillslopes have been predominantly under an annual rotation of wheat (*Triticum aestivum* L.) and fallow (Table 1). This two-crop rotation is with the exception of: a period (1977-1980) of grass (*Psathyrostachys juncea* (Fisch.) Nevski) and a period (1982-1985) of annual wheat on Hillslopes 1 and 2; an annual rotation (1994-2010) of wheat and legume green manure (*Lathyrus sativus* L.) without use of herbicides or mineral fertilizers on Hillslope 1; and an annual rotation (2004-2011) of wheat and pulses (lentils and peas; *Lens culinaris* L. and *Pisum sativum* L.,

respectively) on Hillslope 2. The hillslopes have largely been under conventional tillage practice with a heavy-duty cultivator, with the exception of the period 1993-2011 when Hillslope 2 was switched to no tillage practice, with weed control entirely with herbicides. Unlike the other two hillslopes, Hillslope 3 has had a consistent two-crop rotation and consistent tillage management since 1962.

5    From 1962 to 2011, runoff, runoff nutrient concentration, snowpack characteristics, soil moisture, and soil nutrient concentration were monitored on the hillslopes. The year 2011 marked the final year of regular monitoring of the hillslopes by the Swift Current Research and Development Centre of Agriculture and Agri-food Canada. In 2013 and 2014, monitoring of runoff, snowpack characteristics and soil moisture was undertaken in collaboration with the University of Saskatchewan; these data are presented separately (Appels et al., 2017; Coles and McDonnell, 2018). If regular monitoring of the hillslopes resumes, the data repository will be updated accordingly. Meteorological data from an Environment and

10   Climate Change Canada meteorological station located 390 m to the south-southeast complement the hillslope dataset.

## 3 Previous research with this dataset

These data have proven valuable for the study of a variety of research questions related to the interactions between agricultural practices, climate, hydrology, and nutrient export. Studying the effects of agricultural practices on nutrient export and water quality, Nicholaichuk and Read (1978) showed guideline-exceeding N and P concentrations in runoff from unfertilized hillslopes. Schneider et al. (2019) used the long-term data set to

15   show that the conversion to no-till farming caused an increase in snowmelt-runoff, which subsequently increased nutrient exports. Cessna et al. (2013) found an increased amount of herbicide exported in runoff from conservation tillage hillslopes, due to the increased use of herbicides. McConkey et al. (1997) showed that soil erosion was greatest in April during snowmelt-runoff events over partially-frozen soil, and that erosion occurring during rainfall-runoff events was relatively unimportant. Hydrological process studies at this site include Coles et al. (2018a), who explored the hierarchies of importance of various controls on snowmelt-runoff generation. Appels et al. (2017) and Coles and McDonnell (2018)

20   used single melt season intensive data collection to quantify the spatial patterns of hydrological variables and the generation of snowmelt-runoff connectivity over low-angled terrain. Finally, climate change studies at this site have shown warming winter and spring temperatures, decreasing snowfall amounts and a resultant decrease in snowmelt-runoff amounts, earlier spring runoff timing, and an increase in summer rainfall amounts but no change in rainfall-driven runoff (Cutforth et al., 1999; Coles et al., 2017).

## 4 Available data series

25   The available data are summarized in Table 2 and detailed below.

## 4.1 Runoff

Runoff from the hillslopes is non-continuous, and generated during snowmelt and occasionally during heavy rainfall events. Figure 2 shows the 50-year runoff record for one hillslope (Hillslope 2). Of the 50 years on record, 46 of those years saw measurable flow during spring snowmelt (on at least one of the hillslopes) and 28 years saw measureable hillslope-scale runoff during non-melt rainfall-runoff events (also on at least one of the hillslopes). Because the flumes are so small, the minimum measureable instantaneous flow through the flume is 0.07 L. Assuming a reliable reading would need 30 seconds at that flow rate, this translates to a minimum measureable daily flow of 0.000049 mm/day (Hillslope 1), 0.000045 (Hillslope 2) and 0.000043 mm/day (Hillslope 3). The dataset includes data on daily runoff (mm) and daily peak flow (L s$^{-1}$). Any missing data are shown by an 'NA' and also flagged with an 'm'.

The raised grassed berms prevent runoff from being transferred between the hillslopes. Runoff from the hillslopes is routed through a 0.61 m H-flume (Bos, 1989) at the downstream outlet (north-northeast corner) of each hillslope. The flume state was measured between January 1962 and December 2011 using a Stevens (Portland, Oregon, USA) water level chart recorder in the stilling well of each flume. Since 1994, water level recorder shaft position encoder (Belfort Instrument Co., Baltimore, Maryland, USA) was the primary method to record water level. Daily cumulative runoff amounts (mm) were calculated from these water level measurements using a standard H-flume rating curve (Bos, 1989). The H-flumes were heated in cold weather to prevent icing. Prior to 1993, the flumes were in the open and heated with propane-fueled heaters under the flumes. After 1993, the flume sides and bottoms were electrically heated with resistance heaters and the flumes were in a small building that was also electrically heated (Figure 3). There is no runoff data from March 1969 to November 1970 as the flumes were not monitored. A heavy rainfall event on 14 June 1964 caused flow rates to exceed the flume capacity. Runoff during this event is reported as 'NA' in the dataset, but total daily runoff was estimated to be 72 mm with a peak flow of 60 mm h$^{-1}$ (McConkey et al., 1997). Table 3 provides a summary of the runoff statistics.

## 4.2 Runoff nutrient concentrations

Nutrient concentrations in the runoff were measured on a daily basis during runoff events, from runoff samples taken from the flume at the downstream outlet of each hillslope. Concentration data (in mg L$^{-1}$) are available for nitrate-N (1971-2011), ammoniacal-N (1996-2011), and orthophosphate-P (1971; 1991-2011).

Prior to 1993, on days with appreciable runoff, water samples were collected manually in 0.5-L glass containers at mid-morning (10:00 h ± 30 min) and mid-afternoon (15:00 h ± 30 min). From 1993 onward, 0.5 L samples were collected using an automated water sampler (ISCO 3700

Portable Sampler, Isco, Inc.), previously described by Cessna et al. (2013). The collected water samples were filtered (No. 42, Whatman International filter papers, Maidstone, England) and then analyzed for dissolved $NO_3$-N and orthophosphate-P according to the analytical procedure of Hamm et al. (1970) and for ammonium-nitrogen ($NH_4$-N) according the analytical procedure of Gentry and Willis (1988). The runoff nutrient concentration data are summarised in Table 4.

### 4.3 Snow depth, density, and water equivalent

Snowpack data were determined for each hillslope by manual snow surveys each year between 1965 and 2011. At nine points on each hillslope (as shown in Figure 1), an average snow depth (cm) was determined from multiple locations in the close vicinity of the point with a graduated rod and density (g cm$^{-3}$) was measured once per point as the mass of snow of measured depth taken in in a 7.5-cm diameter core; the mass was determined in the field with a calibrated spring scale. The snow water equivalent (SWE; cm) was calculated from average snow depth multiplied by density. Snow surveys were carried out several times each winter, between January and March. Hillslope-averaged snow depth, density, and SWE were calculated from the nine points for each survey. These data are summarised in Table 5.

### 4.4 Soil moisture

Gravimetric soil moisture content (water fraction by volume of soil) was measured twice each year between 1971 and 2011: in fall prior to freeze-up (sometime in September-November), and in spring following snowmelt (sometime in April-May). The measurements were taken at the same nine points on each hillslope at which snow characteristics were measured (Figure 1), at five depth intervals per point: 0-15 cm, 15-30 cm, 30-60 cm, 60-90 cm, and 90-120 cm. The soil moisture was measured from a subsample of the entire mixed interval and reported for the mid-point of the interval. These gravimetric soil moisture contents were converted to volumetric soil moisture using average bulk density across the hillslopes derived from occasional measurements from mass of soil taken in cores of a known volume and depth. These bulk densities were: 1.26 g cm$^{-3}$ for 0-15 cm, 1.29 g cm$^{-3}$ for 15-30 cm, 1.39 g cm$^{-3}$ for 30-60 cm, 1.54 g cm$^{-3}$ for 60-90 cm, and 1.63 g cm$^{-3}$ for 90-120 cm. Hillslope-averaged soil moisture at each depth was calculated from the nine points. Soil moisture data in the dataset are reported in cm of water for each soil profile depth interval. A summary of the soil moisture aggregated over the entire 0-120 cm soil profile is presented in Table 6.

### 4.5 Soil nutrient concentrations

A subsample from the same samples collected for soil moisture in fall and spring was air-dried and analyzed for $NO_3$-N (1970-1992; 1994-2010), bicarbonate-extractable (Olsen) P (1970-1992; 1994-2010), and ammoniacal-N (1970-1992) (Hamm et al., 1970). Data are reported in the dataset in units of concentration (mg L$^{-1}$) and mass (kg ha$^{-1}$). A summary of the fall soil nutrient concentration data is presented in Table 7.

## 4.6 Meteorology

To complement the hillslope data, meteorological data are available from the Environment and Climate Change Canada meteorological station (station name: Swift Current CDA; Climate ID: 4028060; WMO ID: 71446 and downloadable at http://climate.weather.gc.ca/), 390 m south of the southwest corner of Hillslope 1 and within 1 km of all three hillslopes. These data include daily (1962-present) precipitation (snowfall and rainfall), temperature, wind speed and direction, and snow depth, and hourly (1995-present) temperature, wind speed and direction, and relative humidity.

## 5 Data availability

Data from the three Swift Current hillslopes are available at: https://doi.org/10.23684/hhn5-rz52. This depository includes the runoff, runoff nutrient concentrations, snowpack, soil moisture, soil nutrient concentrations, and crop and tillage data. Herbicide and sediment concentration data are not included in this dataset. We have also made available the digital elevation data for the site, which can be accessed at: https://doi.org/10.20383/101.0117 (Coles et al., 2018b). The Environment and Climate Change Canada meteorological data are available at: http://climate.weather.gc.ca/.

## 6 Summary

This paper has presented details of a long-term (1962-2013) dataset from three 5 ha research hillslopes at the Swift Current Research and Development Centre (of Agriculture and Agri-food Canada) in southern Saskatchewan on the northern Great Plains. The hillslopes are seasonally-frozen and under agricultural management. Runoff from the hillslopes is non-continuous and is dominated by spring snowmelt. The dataset includes runoff, runoff nutrient concentrations, snowpack depth, density and water equivalent, soil moisture, soil nutrient concentrations, and crop and tillage data. Digital elevation data are also available for the three hillslopes at a 2 m horizontal resolution, and, for one of the hillslopes (Hillslope 2), at a 0.25 m horizontal resolution. A nearby Environment and Climate Change standard meteorological station provides a dataset of precipitation, temperature and other climate conditions at a daily (1962 onwards) and hourly (1995 onwards) resolution. Data from the three hillslopes and the meteorological station are now available online. This rich dataset is a valuable source for hydrological response analysis and model formulation and calibration, within the context of climate and land management change.

**Acknowledgements**

This hillslope dataset exists thanks to the hard work of many Agriculture and Agri-food Canada researchers, technicians, and students over the last six decades. Don Reimer and Marty Peru are thanked especially for their dedicated technical service to the hillslopes from 1971 to 1994, and from 1995 to 2011, respectively. We thank Warren Helgason, Michael Solohub and Amber Peterson (University of Saskatchewan) and Mark Russell, Cuyler Onclin, and Samson Mengistu (Environment and Climate Change Canada) for their surveying efforts to create the DEMs. We thank Willemijn Appels for her contributions to the dataset, and Chris Spence and Charles Maule for their earlier reviews of this manuscript.

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

## Figures and tables

**Table 1.** Crop rotation† (and tillage) history of the three hillslopes.

| Time Period | Hillslope 1 | Hillslope 2 | Hillslope 3 |
|---|---|---|---|
| 1962-1976 | F-W (blade and conv‡ till) | W-F (blade and conv till) | F-W (disc and conv till) |
| 1977-1980 | Unfertilized grass for hay (no-till) | Unfertilized grass for hay (no-till) | F-W (disc and conv till) |
| 1981-1986 | F-W-W-W-F (conv till and cultivator) | F-W-W-W-F (conv till and cultivator) | F-W (conv till and cultivator) |
| 1987-1992 | W-F (conv till and cultivator) | W-F (conv till and cultivator) | W-F (conv till and cultivator) |
| 1993-2002 | W-GM (conv till and incorporated) | W-F (no-till) | W-F (conv till and cultivator) |
| 2003-2011 | W-GM (conv till and incorporated) | W-LP (no-till) | W-F (conv till and cultivator) |

†F - fallow, W - spring wheat (*Triticum aestivum* L.), grass (*Psathyrostachys juncea* (Fisch.) Nevski), GM - green manure chickling vetch (*Lathyrus sativus*), LP - pulse crop, lentil (*Lens culinaris* L.) or field pea (*Pisum sativum* L.) in alternating pulse crop years,

5    ‡conv till – conventional tillage with cultivator including harrows or rod-weeder used in last operation before seeding with hoe drill.

**Table 2.** Summary of data available and years recorded.

| Data | Years |
|---|---|
| Runoff | |
| - Daily total (mm) | - 1962-1969; 1971-2011 |
| - Daily peak flow (L s$^{-1}$) | - 1962-1969; 1971-2011 |
| Runoff nutrient concentrations (mg L$^{-1}$) | |
| - Nitrate-N (NO$_3$-N) | - 1971-2011 |
| - Ammoniacal-N (NH$_4$-N) | - 1996-2011 |
| - Orthophosphate-P (PO$_4^{3-}$) | - 1971; 1991-2011 |
| Snowpack | |
| - Depth (cm) | - 1965-2011 (1-6 surveys per year) |
| - Density (mg cm$^{-3}$) | - 1965-2011 (1-6 surveys per year) |
| - Snow water equivalent (cm) | - 1965-2011 (1-6 surveys per year) |
| Soil moisture content (cm) | |
| - 0-15, 15-30, 30-60, 60-90, 90-120 cm | - 1971-2011 (April and October) |
| Soil nutrient (NO$_3$-N, NH$_4$-N, and PO$_4^{3-}$) concentrations (mg L$^{-1}$) | |
| - 0-15 cm | - 1970-1992 (April and October) |
| - 0-15, 15-30, 30-60, 60-90, 90-120 cm | - 1993-2010 (April and October) (NO$_3$-N and PO$_4^{3-}$only) |
| Crop type and management | - 1962-2011 |
| Meteorology (ECCC data): | |
| - Daily: maximum temperature, minimum temperature, rain, snow, precipitation, snow depth on ground, wind speed and direction | - 1962-2011+ (ongoing) |
| - Hourly: temperature, dew point, relative humidity, wind speed and direction | - 1995-2011+ (ongoing) |

**Table 3.** Summary statistics of runoff data.

| | Hillslope 1 | Hillslope 2 | Hillslope 3 |
|---|---|---|---|
| Mean annual runoff (± std) (mm) | 25.4 ± 33.9 | 34.9 ± 38.9 | 20.5 ± 21.8 |
| **Maximum daily runoff (mm)** | | | |
| - Annual (Jan-Dec) | 45.2 (9 Apr 1965) | 48.7 (25 Feb 1986) | 28.5 (11 Apr 1974) |
| - Spring (Feb-Apr) | 45.2 (9 Apr 1965) | 48.7 (25 Feb 1986) | 28.5 (11 Apr 1974) |
| - Summer (Jun-Aug) | 8.9 (16 June 1995) | 12.1 (1 July 1982) | 9.2 (3 Jun 1976) |
| **Mean (± std) daily runoff (mm)** | | | |
| - Annual (Jan-Dec) | 0.1 ± 0.9 | 0.1 ± 1.1 | 0.1 ± 0.7 |
| - Freshet (Feb-Apr) | 0.3 ± 1.8 | 0.4 ± 2.2 | 0.2 ± 1.3 |
| - Summer (Jun-Aug) | 0.0 ± 0.2 | 0.0 ± 0.2 | 0.0 ± 0.3 |
| **Maximum daily peak flow (L s$^{-1}$)** | | | |
| - Annual (Jan-Dec) | 119.2 (1 Jul 1982) | 233.3 (1 Jul 1982) | 132.2 (1 Jul 1982) |
| - Freshet (Feb-Apr) | 85.8 (8 Apr 1965) | 115.3 (9 Apr 1965) | 64.0 (11 Apr 1974) |
| - Summer (Jun-Aug) | 119.2 (1 Jul 1982) | 233.3 (1 Jul 1982) | 132.2 (1 Jul 1982) |
| **Mean (± std) daily peak flow (L s$^{-1}$)** | | | |
| - Annual (Jan-Dec) | 0.2 ± 2.3 | 0.3 ± 3.3 | 0.2 ± 2.6 |
| - Spring (Feb-Apr) | 0.6 ± 3.7 | 0.8 ± 5.0 | 0.5 ± 3.3 |
| - Summer (Jun-Aug) | 0.1 ± 2.6 | 0.2 ± 4.2 | 0.2 ± 3.9 |

**Table 4.** Summary statistics of runoff nutrient concentration data.

| | Hillslope 1 | Hillslope 2 | Hillslope 3 |
|---|---|---|---|
| Nitrate-N ($NO_3$-N) (mg $L^{-1}$) | | | |
| - Minimum | 0.01 (16 Mar 1975) | 0.01 (16 Mar 1975) | 0.0 (12 Mar 1996) |
| - Maximum | 9.9 (17 Mar 2011) | 8.1 (22 Mar 2011) | 5.7 (18 Mar 2011) |
| - Mean ± std | 1.3 ± 1.6 | 0.8 ± 1.0 | 1.0 ± 1.1 |
| Ammoniacal-N ($NH_4$-N) (mg $L^{-1}$) | | | |
| - Minimum | 0.0 (12 Mar 1996) | 0.0 (12 Mar 1996) | 0.0 (11 Mar 1996) |
| - Maximum | 4.6 (23 Mar 2006) | 4.5 (4 Feb 2005) | 3.3 (1 Feb 2005) |
| - Mean ± std | 0.4 ± 0.8 | 0.4 ± 0.6 | 0.3 ± 0.5 |
| Orthophosphate-P ($PO_4^{3-}$) (mg $L^{-1}$) | | | |
| - Minimum | 0.0 (21 Feb 1996) | 0.0 (20 Feb 1996) | 0.0 (22 Feb 1996) |
| - Maximum | 11.8 (24 Jun 1991) | 1.7 (22 Jul 2004) | 1.5 (15 Mar 2011) |
| - Mean ± std | 0.5 ± 1.5 | 0.3 ± 0.2 | 0.3 ± 0.3 |

20

25

**Table 5.** Summary statistics of hillslope-averaged snowpack data (only using data from the last survey before the end of winter).

| | Hillslope 1 | Hillslope 2 | Hillslope 3 |
|---|---|---|---|
| Snowpack depth (cm) | | | |
| - Minimum | 0.0 (1968, 1970) | 0.0 (1970) | 0.0 (multiple) |
| - Maximum | 38.4 (1978) | 37.8 (1978) | 27.9 (1967) |
| - Mean ± std | 12.4 ± 8.06 | 15.3 ± 8.09 | 11.0 ± 7.68 |
| | | | |
| Density (g cm$^{-3}$) | | | |
| - Minimum | 0.0 (1970) | 0.0 (1968, 1970) | 0.0 (multiple) |
| - Maximum | 0.5 (1966) | 0.4 (1969, 1971) | 0.4 (1966) |
| - Mean ± std | 0.3 ± 0.1 | 0.3 ± 0.1 | 0.2 ± 0.1 |
| | | | |
| Snow water equivalent (cm) | | | |
| - Minimum | 0.0 (1968, 1970) | 0.0 (1968, 1970) | 0.0 (multiple) |
| - Maximum | 11.5 (1978) | 12.1 (1978) | 7.8 (1967) |
| - Mean ± std | 3.2 ± 2.5 | 4.2 ± 2.7 | 2.8 ± 2.1 |

20

**Table 6.** Summary statistics of soil moisture data (for simplicity, data are aggregated over the five depth intervals and presented for the entire 0-120 cm soil profile, however data are recorded separately by depth interval in the dataset).

|  | **Hillslope 1** | **Hillslope 2** | **Hillslope 3** |
|---|---|---|---|
| Fall soil moisture (cm) | | | |
| - Minimum | 7.77 (2005) | 9.48 (2004) | 7.23 (2005) |
| - Maximum | 37.7 (1993) | 32.6 (1996) | 33.2 (1992) |
| - Mean ± std | 20.4 ± 5.72 | 20.6 ± 6.27 | 22.1 ± 5.81 |
| | | | |
| Spring soil moisture (cm) | | | |
| - Minimum | 7.87 (2008) | 11.1 (2008) | 10.2 (2008) |
| - Maximum | 34.6 (1995) | 40.8 (1995) | 36.8 (1997) |
| - Mean ± std | 22.1 ± 4.88 | 22.8 ± 5.53 | 23.6 ± 4.61 |

**Table 7.** Summary statistics of soil nutrient concentration data.

| | Hillslope 1 | Hillslope 2 | Hillslope 3 |
|---|---|---|---|
| Nitrate-N ($NO_3$-N) (mg $L^{-1}$) | | | |
| - Minimum | 0.4 (1995) | 0.3 (1997) | 0.3 (1997) |
| - Maximum | 19.0 (1981, 1982) | 17.7 (2009) | 23.0 (1971) |
| - Mean ± std | 4.4 ± 3.9 | 3.5 ± 3.5 | 3.4 ± 3.4 |
| Ammoniacal-N ($NH_4$-N) (mg $L^{-1}$) | | | |
| - Minimum | 3.3 (1974) | 4.1 (1981, 1982) | 4.1 (1974) |
| - Maximum | 10.4 (1984) | 11.5 (1984) | 14.2 (1971) |
| - Mean ± std | 5.6 ± 2.1 | 5.9 ± 2.5 | 6.8 ± 3.4 |
| Orthophosphate-P ($PO_4^{3-}$) (mg $L^{-1}$) | | | |
| - Minimum | 0.6 (2004) | 0.3 (2004) | 0.6 (1997) |
| - Maximum | 15.1 (1989) | 23.0 (2009) | 22.6 (1971) |
| - Mean ± std | 5.8 ± 4.4 | 6.2 ± 5.8 | 7.0 ± 3.4 |

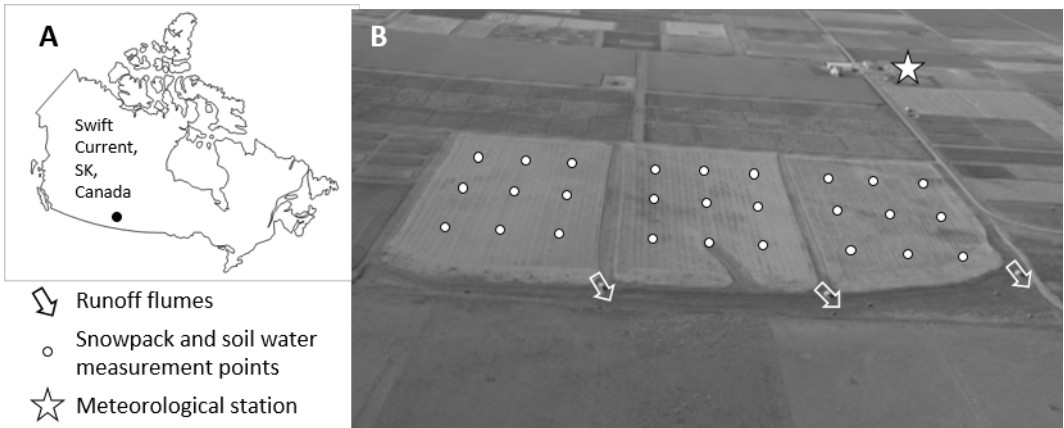

**Figure 1.** A) Location of the Swift Current hillslopes in Canada. B) Aerial photograph (facing south) of the Swift Current hillslopes (from right to left: Hillslope 1, Hillslope 2, Hillslope 3), taken in a year when wheat was grown. The runoff flumes, measurement locations for snowpack and soil moisture, and the meteorological station are indicated. Figure modified, with permission, from Cessna et al. (2013) and Coles et al. (2017).

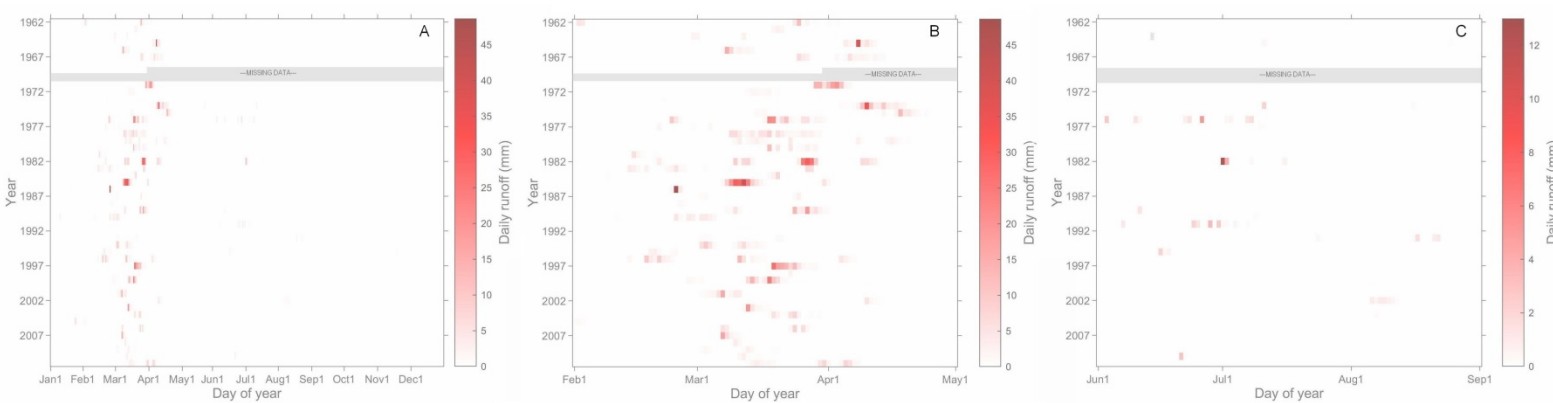

**Figure 2.** Daily runoff (mm) for the 50-year period on Hillslope 2. Panel A shows the full calendar year for the 50-year period (January 1st 1962 to December 31st 2011). Panel B shows just the spring period (here, February 1st to May 1st), for increased clarity of the spring snowmelt-runoff data. Panel C shows just the summer period (here, June 1st to September 1st), for increased clarity of the summer rainfall-runoff data.

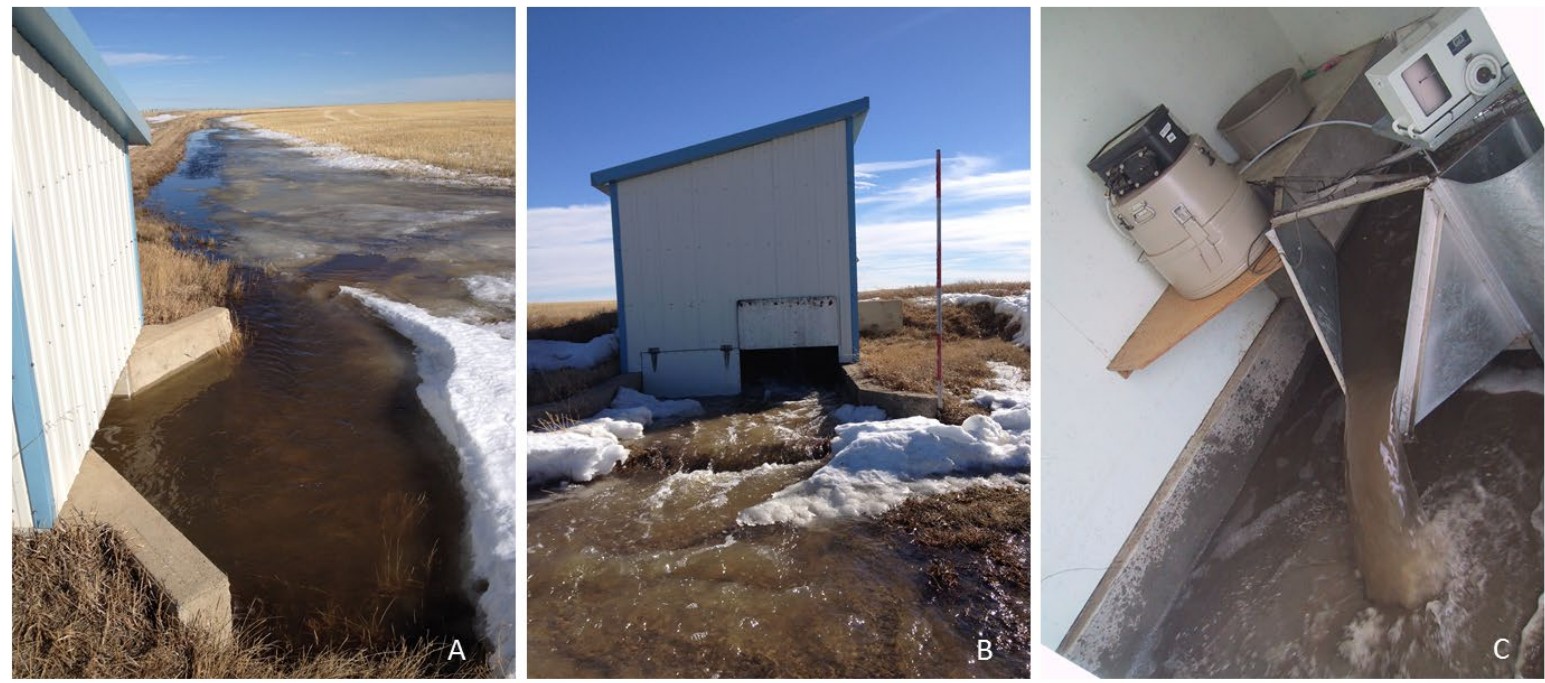

**Figure 3.** Photographs of the runoff shed at Hillslope 2. A) Facing east, at the inflow side of the shed (taken March 2014). B) Facing south, at the outflow side of the shed (taken March 2014). C) Inside the shed, showing a 0.61m H flume, a Stevens water level chart recorder, and an automated ISCO 3700 Portable Sampler.

