# Peer review of "Fifty years of recorded hillslope runoff on seasonally-frozen ground: The Swift Current, Saskatchewan, Canada dataset"

_Earth System Science Data, 2018_

## Referee Comment (RC1) · Spence (Referee) · 17 Jan 2019

In this paper, the authors summarize a hydrological and chemistry dataset from a set of experimental hillslopes in Saskatchewan, Canada. It is a very nice dataset, which deserves to be catalogued and preserved. Its value is certainly enhanced by the long period of record. The data are easily accessible from Government of Canada open data websites. Upon reading the paper, I felt like there should be more description of the data, and more information on the methods used to collect it. As it is now, the paper does not provide enough information, particularly of the latter, for new users of the data to maximize its usage. My comments, both major and minor, are below.

[Figure]

Page 1 Line 10: perhaps say "nutrient flux (or concentration or export)"

Page 1 Line 13: Perhaps pick one of "edge of field" or "hillslope" and stick with it throughout the paper.

Page 1: Line 20: The digital elevation data that are mentioned here should be introduced earlier in the abstract.

There are very minor grammatical errors throughout, the kind that can perhaps be addressed by a copy editor at the end of the review period. However, these should be fixed during the next revision via a thorough proof read.

Page 2 Line 2: I have never liked the phrase "to our knowledge" because they always reveal that the authors have not done their homework. For instance, the Experimental Lakes Area have been documenting runoff from several hillslopes since the 1970s. There are not many research hillslopes over frozen ground, but please do not perpetuate the idea that this is the only dataset that exists. Turkey Lakes, Trail Valley Creek, Wolf Creek, McMaster Basin; these are all places that have comparable data. Furthermore, aren't the Swift Current hillslopes in existence? I suggest rephrasing non-existent to rare or uncommon. What is distinct is the period of record. Long, very long.

Page 2 Line 12: It has not been determined how applicable the results found at Swift Current are in other climates and landscapes. Perhaps temper the statement by saying "in this landscape or "in dry agroecosystems".

Page 3 Line 2: It is unclear how the hillslopes were surveyed with the Leica instruments. Could the authors please provide information on the projection, datum and accuracy of the elevation data. Also, could you please describe the data. What are the mean elevation, slopes and relief, for instance?

Title of Section 3: Could I suggest rephrasing this? "Previous research"; "Prior research" Just some suggestions.

Page 3 Line 26: A lead in sentence would help here. "These data have proven valuable for the study of a variety of research relevant to the interactions among climate, hydrology, material export and agricultural practices.

Page 3 Line 26: Perhaps "long term" is a phrase that is not required throughout the paper.

Page 3 Line 27: Please expand on what each of these studies found.

Page 4 Line 2: Perhaps rephrase to: "The data have also been used to . . .."

Page 4 Line 5: The structure of this paragraph seems to jump all over. Categorize. Perhaps discuss 1) hydrological process studies; 2) effect of different ag practices; 3) material export a) erosion, b) nutrients, c) water quality; and 4) climate change.

Page 4 Line 11: Please describe the data and provide numbers. Range, average annual runoff, standard deviations, peaks, annual yields, etc. would all be interesting for the reader.

Page 4 Line 18: How were these heated? The authors should mention that the flumes are inside sheds. Maybe even provide pictures. This all helps people understand how the data were collected.

Page 4 Line 18: "The only event to exceed flume capacity was generated by a heavy rainfall event on 14 June 1964." Please describe explicitly how this gap was filled. The paper needs to stand on its own, and not lean on citations to others that have described the methodologies used to collect the data. That is one of the major points of a data paper. Centralized information.

Page 4 Line 21: Please describe the nutrient export data and provide some plots illustrating the data.

Page 5: For each of the datasets, please describe the data and provide illustrations of them Section 4.3: "Snowpack characteristics were measured". . .. Please describe the

equipment used during the snow surveys. Did this equipment ever change?

Page 5 Line 7: "The measurements were taken at the same nine points on each hillslope at which snow traits were measured (Figure 1).

Page 5 Line 9: "volumetric soil moisture"

Page 5 Line 10: What is the soil core method?

Section 4.6: Because these are publically available data from an operational national climate network that were not collected by the authors, I do not think they should be included.

Page 5 Line 23: There is no description of the agricultural practices data, or it is too brief. This is an important detail, which should be highlighted more with some kind of time series plot.

Acknowledgements: Willemijn Appels?

Figure 2: I am not sure this is the best way to present this data. The short events are really hard to resolve, especially once it gets into a journal. Contours, if you wish to use this kind of plot? Maybe split the x-axis, because nothing ever happens between August and February.
* * *

---

## Referee Comment (RC2) · Charles Maule (Referee) · 11 Feb 2019

GENERAL COMMENTS This manuscript presents data for three agricultural fields that were monitored for snowpack characteristics (depth, SWE), pre- freezeup and post-melt soil moisture, daily runoff flows and nutrients, and climate from 1962 to 2011. The fields represent dryland farming in the semi-arid portion of the Canadian Prairies. The data set is unique in that it provides longterm continuous and fairly comprehensive climate, soil, and snowpack data that control snowmelt runoff.  This dataset will prove very useful to researchers within snow hydrology and water quality, especially as it represents 50 years of documented data and management methods.  With the exception

of minor errors the manuscript is well written and structured. The data sets are also well organized and easily accessible.

SPECIFIC COMMENTS Should indicate whether or not the plots are still active after 2011. If data is still being collected should be a sentence about updating the data files. 2,25: is any of the sediment data available? The plots were set up to investigate soil erodibility and reported upon by McConkey et al (1997) and Nicholaichuk and Read (1978) (4,1).

TECHNICAL CORRECTIONS 1,12: specify what is meant by 'snowpack'? 1,17: a period is needed between '2011' and 'Gravimetric'. 1,18: are the data sets 'summaries' or the complete data sets? 4,10: as the runoff data set reports values as small as 0.01 mm/d with flag indicators of 'good observations', what minimum value is considered measurable flow? 4,11: does the "on at least one of the hillslopes" apply to the spring snowmelt as well? 3,9-11: reference for the two sentences covering these lines. 4,18: "No runoff was measured during 1970" could be read as no measurable runoff occurred. From the data set it appears that 'The H flumes were not operational (or not measured?) from March 1969 through to end of November 1970 thus no data is reported.' Is this the only instance that the flume was not operational/not measured? 4,19: should clarify in the text that the value is reported as NA in the dataset and estimated values can be found in McConkey et al (1997). 4,22: to maintain consistency with data units should use the terms 'nitrate-N', 'ammoniacal-N', and 'phosphate-P'. 4,23: Cessna et al (2013) refers to a herbicide paper that has no analysis description of nutrients. 5,6: as fall sample dates also occur in September and November perhaps just state samples taken in fall prior to freezeup and 'in spring following snowmelt' (as samples were also taken in May ). 5,18: as hourly data does not include precipitation or snow depth perhaps reword this sentence so it is clear. 6,7: 'snowpack' characteristics? Table 1: hourly meteorology data? To what set of data does the '1994-2011+ (ongoing)' refer to? Figure 2: some of the light (less than 10 mm) daily runoff values are very difficult to discern. Although it is realized that the figure is for example purposes

only it is important to note some of the occurrences of runoff, especially during the warm season. Suggest that the color shading start at a higher value of blue or that two colors are used, not white and/or a note about what value is considered 'measurable' on this figure.

DATA REVIEW only refers to the English web links, except where noted. 1962-2011 DATA ON SOILS, RUNOFF, AND SNOW. All data was checked for outliers. Phosphate-P: Data: Watershed 1 1991 has values an order of magnitude higher than all other values. Runoff: should mention in the manuscript that daily peak flow values are also available as this information can be very useful. Runoff data: 'NA' occurs in the Runoff and Peak columns not the Flag Indicator column. The flag for the 'NA' values is 'm'. Snow Water Equivalent: snow Density units are listed as 'Mg/cm3'. Should be 'g/cm3' according to the format given in the data file. French version 'mg/cm3' should also be 'g/cm3'. Soil Moisture Content: depths are wrong for d60_d90 and d90_d120. Ok for French version. Within dataset some values for the deeper intervals are very low (for soils with clay contents greater than 20%) and must be due to sand pockets. Soil Nitrate Phosphorus: Dictionary: Depth intervals, some errors in both English and French versions. Soil Nutrients: English Dictionary: French version comes up. Soil Nutrients Dataset: years and depth should be in same format of other soil files (eg Soil_Nitrate_Phosphorous). Watershed Management: Dictionary: System, n is missing from word 'rotation'. Watershed Management Data: under 'System' Column there is 'wheat-GM fallow'. This should be 'wheat-green manure fallow' according to the Dictionary and to the 'Management Systems Details' data set. Some cells are lacking information where it is expected; for example: Column D (Previous Crop) rows 33, 127-133; Column E rows 51, 99, 151. METEOROLOGICAL DATA. Dates of start and finish and parameters were noted for daily and hourly data. Data was only spot reviewed. No issues were noted. DIGITAL ELEVATION DATA. The description and README.txt file was read. Data was not reviewed. Description file ; 2nd last sentence; "(random toughness)"?

---

## Author Comment (AC1) · 6 Apr 2019

Author responses to referee comments on "Fifty years of recorded hillslope runoff on seasonally-frozen ground: The Swift Current, Saskatchewan, Canada dataset" by Anna E. Coles et al.

4 April 2019

In this comment, we (the authors) respond to each of the comments made by the two referees – Chris Spence and Charles Maule – and detail the associated changes to the manuscript. Thank you to Drs Spence and Maule for their critical and constructive

feedback. Original referee comments are denoted by RC, author responses as AC, and manuscript changes as MC.

RC1: Spence

RC1.1 In this paper, the authors summarize a hydrological and chemistry dataset from a set of experimental hillslopes in Saskatchewan, Canada. It is a very nice dataset, which deserves to be catalogued and preserved. Its value is certainly enhanced by the long period of record. The data are easily accessible from Government of Canada open data websites. Upon reading the paper, I felt like there should be more description of the data, and more information on the methods used to collect it. As it is now, the paper does not provide enough information, particularly of the latter, for new users of the data to maximize its usage. My comments, both major and minor, are below. AC1.1 Thank you for your comments. We couldn't agree more that this dataset deserves to be preserved and used for future studies. We agree in hindsight that there should have been more information on methods and descriptive statistics of the data. We respond to your individual comments pertaining to this and other aspects, below. MC1.1 We have edited the revised manuscript to address all of your comments. Specific changes detailed below.

RC1.2 Page 1 Line 10: perhaps say "nutrient flux (or concentration or export)" AC1.1 Agreed. MC1.1 Edited to "nutrient concentrations"

RC1.3 Page 1 Line 13: Perhaps pick one of "edge of field" or "hillslope" and stick with it throughout the paper. AC1.2 Agreed. MC1.2 There was one occurrence of "edge-of-field" and one of "field" in the manuscript. Changed these to "hillslope" to be consistent with the rest of the manuscript.

RC1.4 Page 1: Line 20: The digital elevation data that are mentioned here should be introduced earlier in the abstract. AC1.4 Agreed. MC1.4 Added a line earlier in the abstract to say that digital elevation data are available for the three hillslopes at a 2 m resolution, and also at a 0.25 m resolution for one of the hillslopes (Hillslope 2)

RC1.5 There are very minor grammatical errors throughout, the kind that can perhaps be addressed by a copy editor at the end of the review period. However, these should be fixed during the next revision via a thorough proof read. AC1.5 Apologies for missing these errors. We have made a thorough proofread. MC1.5 We have made grammatical corrections throughout the manuscript.

RC1.6 Page 2 Line 2: I have never liked the phrase "to our knowledge" because they always reveal that the authors have not done their homework. For instance, the Experimental Lakes Area have been documenting runoff from several hillslopes since the 1970s. There are not many research hillslopes over frozen ground, but please do not perpetuate the idea that this is the only dataset that exists. Turkey Lakes, Trail Valley Creek, Wolf Creek, McMaster Basin; these are all places that have comparable data. Furthermore, aren't the Swift Current hillslopes in existence? I suggest rephrasing non-existent to rare or uncommon. What is distinct is the period of record. Long, very long. AC1.6 I (Anna) really appreciate this comment, thank you. As well as the adjustments in this manuscript (next comment), I will be mindful of being much more thorough not using phrasing like this in the future. MC1.6 We have modified the manuscript to remove "to our knowledge" and have edited "non-existent" to "uncommon".

RC1.7 Page 2 Line 12: It has not been determined how applicable the results found at Swift Current are in other climates and landscapes. Perhaps temper the statement by saying "in this landscape or "in dry agroecosystems". AC1.7 Agreed MC1.7 Edited to "in terms of land use change in dry agroecosystems"

RC1.8 Page 3 Line 2: It is unclear how the hillslopes were surveyed with the Leica instruments. Could the authors please provide information on the projection, datum and accuracy of the elevation data. Also, could you please describe the data. What are the mean elevation, slopes and relief, for instance? AC 1.8 Agreed that this information is necessary MC 1.8 Edited the manuscript to incorporate information on the survey equipment and the resultant data. We have also added descriptive statistics on the topographic characteristics.
RC1.9 Title of Section 3: Could I suggest rephrasing this? "Previous research"; "Prior research" Just some suggestions. AC 1.9 Agree that modification of the existing title ("Work with the data to date") would be an improvement, while still making sure that it is clear that it we are talking about research using this dataset. MC 1.9 Edited to "Previous research with this dataset"

RC1.10 Page 3 Line 26: A lead in sentence would help here. "These data have proven valuable for the study of a variety of research relevant to the interactions among climate, hydrology, material export and agricultural practices. AC 1.10Thank you, yes this would improve this paragraph. MC 1.10 Edited to include a lead in sentence.

RC1.11 Page 3 Line 26: Perhaps "long term" is a phrase that is not required throughout the paper. AC1.11 You are quite right, we were over-doing it with "long-term" MC1.11 Removed four "long-term"s, where they were not needed.

RC1.12 Page 3 Line 27: Please expand on what each of these studies found. AC1.12 Agreed, this is good information to include. MC1.12 We have provided information on those studies' key findings.

RC1.13 Page 4 Line 2: Perhaps rephrase to: "The data have also been used to . . .." AC1.13 Agreed MC1.13 Edited the manuscript to this.

RC1.14 Page 4 Line 5: The structure of this paragraph seems to jump all over. Categorize. Perhaps discuss 1) hydrological process studies; 2) effect of different ag practices; 3) material export a) erosion, b) nutrients, c) water quality; and 4) climate change. AC1.14 Agreed that this paragraph is a little all over the place. MC1.14 We have edited this paragraph to improve flow.

RC1.15 Page 4 Line 11: Please describe the data and provide numbers. Range, average annual runoff, standard deviations, peaks, annual yields, etc. would all be interesting for the reader. AC1.15 Agreed. MC1.15 Edited the manuscript to include runoff statistics describing the data.
RC1.16 Page 4 Line 18: How were these heated? The authors should mention that the flumes are inside sheds. Maybe even provide pictures. This all helps people understand how the data were collected. AC1.16 Agreed, thanks for this suggestion MC1.16 Edited the manuscript to add information about the sheds and described how they are set up and heated. We have added a photo of the outside and inside of one of the sheds (at Hillslope 2).

RC1.17 Page 4 Line 18: "The only event to exceed flume capacity was generated by a heavy rainfall event on 14 June 1964." Please describe explicitly how this gap was filled. The paper needs to stand on its own, and not lean on citations to others that have described the methodologies used to collect the data. That is one of the major points of a data paper. Centralized information. AC1.17 Thank you, we agree with your point here MC1.17 We have added in the methodology used to estimate the 1964 flow. We have still referenced the original paper of course, but have explicitly included that information rather than referring the reader to that paper to find it. We have done the same for the sample collection protocols and analysis techniques for the runoff and soil nutrient concentrations.

RC1.18 Page 4 Line 21: Please describe the nutrient export data and provide some plots illustrating the data. AC1.18 Agreed that this information is lacking and that more figures illustrating the data (not just the runoff data) are necessary. MC1.18 Have added more information about the nutrient concentration data in the section "Runoff nutrient concentrations" and created time series plots illustrating the nutrient data.

RC1.19 Page 5: For each of the datasets, please describe the data and provide illustrations of them Section 4.3: "Snowpack characteristics were measured". . .. Please describe the equipment used during the snow surveys. Did this equipment ever change? AC1.19 Yes, like the above comment, we agree that more data descriptions and illustrations are necessary. MC1.19 We have provided more description of the data in the "Snow depth, density and water equivalent", "Soil moisture" and "Soil nutrient concentrations" sections, and created accompanying figures illustrating each of the data
types. We have written in information on the snow survey equipment used.

RC1.20 Page 5 Line 7: "The measurements were taken at the same nine points on each hillslope at which snow traits were measured (Figure 1). AC1.20 Agreed, this is a good clarification. MC1.20 Edited the manuscript to that (but used "characteristics" instead of "traits").

RC1.21 Page 5 Line 9: "volumetric soil moisture" AC1.21Yes MC1.21 Edited to that.

RC1.22 Page 5 Line 10: What is the soil core method? AC1.22 Strange use of words there, sorry for the confusion. We mean standard method of bulk density measurement, by taking a soil sample and weighing its wet and dry weight to determine bulk density. MC1.22 Edited the manuscript to describe the standard method of bulk density measurement, with appropriate reference.

RC1.23 Section 4.6: Because these are publically available data from an operational national climate network that were not collected by the authors, I do not think they should be included. AC1.23 We disagree. We are not pretending that these are our collected data. We are merely showing the reader/user of the data the location of the best/most relevant meteorological data that would naturally be have to used in any analysis of the snowmelt- or rainfall-runoff data that we do present. MC1.23 No changes made.

RC1.24 Page 5 Line 23: There is no description of the agricultural practices data, or it is too brief. This is an important detail, which should be highlighted more with some kind of time series plot. AC1.24 Agreed. There should be more information on this. MC1.24 Edited the manuscript to include more detailed description of the crop rotations, tillage management, and machinery used.

RC1.25 Acknowledgements: Willemijn Appels? AC1.25 Absolutely, quite right, thank you. MC1.25 Acknowledgement of Willemijn Appels' contribution.

Figure 2: I am not sure this is the best way to present this data. The short events are

really hard to resolve, especially once it gets into a journal. Contours, if you wish to use this kind of plot? Maybe split the x-axis, because nothing ever happens between August and February. AC 2.18 Thanks for this feedback. Both you and Referee 2 thought the same, and I agree this figure can be much improved. MC 2.18 Created a new figure with improved, dual colour shading to see the small runoff events. Also created an inset of just spring snowmelt events and an inset of just summer events, to better see those events. Di not truncate the axis as there are an occasional event in late summer, fall and winter (that are now clearer to see due to the changed colour shading).

RC2: Charles Maule

RC2.1 GENERAL COMMENTS This manuscript presents data for three agricultural fields that were monitored for snowpack characteristics (depth, SWE), pre- freezeup and postmelt soil moisture, daily runoff flows and nutrients, and climate from 1962 to 2011. The fields represent dryland farming in the semi-arid portion of the Canadian Prairies. The data set is unique in that it provides longterm continuous and fairly comprehensive climate, soil, and snowpack data that control snowmelt runoff. This dataset will prove very useful to researchers within snow hydrology and water quality, especially as it represents 50 years of documented data and management methods. With the exception of minor errors the manuscript is well written and structured. The data sets are also well organized and easily accessible. AC2.1 Thank you for your time reviewing this manuscript. We address your specific comments below. MC2.1 N/A

SPECIFIC COMMENTS

RC2.2 Should indicate whether or not the plots are still active after 2011. If data is still being collected should be a sentence about updating the data files. AC2.2 The plots are not currently active, but might be in the future. MC2.2 Added some lines to say that if regular monitoring of the hillslopes resumes, the data repository will be updated accordingly.

RC2.3 2,25: is any of the sediment data available? The plots were set up to investigate soil erodibility and reported upon by McConkey et al (1997) and Nicholaichuk and Read (1978) (4,1). AC2.3 Sediment data is not included in this dataset. MC2.3 Added a line to clarify this.

TECHNICAL CORRECTIONS

RC2.4 1,12: specify what is meant by 'snowpack'? AC2.4 Thanks, yes, we mean snowpack depth, density and water equivalent. MC2.4 Edited the manuscript to say that.

RC2.5 1,17: a period is needed between '2011' and 'Gravimetric'. AC2.5 Done MC2.5 Done

RC2.6 1,18: are the data sets 'summaries' or the complete data sets? AC2.6 Thanks for picking that up. Incorrect use of "summarize"! The data sets are complete, and are not summaries. MC2.6 Edited to "We provide these data" instead of "We summarize these data"

RC2.7 4,10: as the runoff data set reports values as small as 0.01 mm/d with flag indicators of 'good observations', what minimum value is considered measurable flow? AC2.7 Because the flumes are so small, the minimum measureable instantaneous flow through the flume is 0.07 L. Assuming a reliable reading would need 30 seconds at that flow rate, this translates to a minimum measureable daily flow of 0.000049 mm/day (Hillslope 1), 0.000045 (Hillslope 2) and 0.000043 mm/day (Hillslope 3). MC2.7 Edited the manuscript to include this information.

RC2.8 4,11: does the "on at least one of the hillslopes" apply to the spring snowmelt as well? AC2.8 Yes MC2.8 Edited to clarify that.

RC2.9 3,9-11: reference for the two sentences covering these lines. AC2.9 Added the reference MC2.9 Added the reference

RC2.10 4,18: "No runoff was measured during 1970" could be read as no measurable

runoff occurred. From the data set it appears that 'The H flumes were not operational (or not measured?) from March 1969 through to end of November 1970 thus no data is reported.' Is this the only instance that the flume was not operational/not measured? AC2.10 No runoff was measured because the H flumes were not operational, therefore during that period we neither know if runoff occurred nor, if it did, how much runoff there was. That was the only instance of the flumes being non-operational. MC2.10 Edited the text to clarify that.

RC2.11 4,19: should clarify in the text that the value is reported as NA in the dataset and estimated values can be found in McConkey et al (1997). AC2.11 Good point, we clarify that. MC2.11 Edited to clarify that.

RC2.12 4,22: to maintain consistency with data units should use the terms 'nitrate-N', 'ammoniacal-N', and 'phosphate-P'. AC2.12 Thanks, this is a good point MC2.12 Edited all references to nutrients in the text to 'nitrate-N', 'ammoniacal-N', and 'phosphate-P'

RC2.13 4,23: Cessna et al (2013) refers to a herbicide paper that has no analysis description of nutrients. AC2.13 This should be a different paper, sorry for that mistake and thanks for catching it. MC2.13 Added correct reference to manuscript.

RC2.14 5,6: as fall sample dates also occur in September and November perhaps just state samples taken in fall prior to freezeup and 'in spring following snowmelt' (as samples were also taken in May ). AC2.14 Right, this needs changing. MC2.14 Edited to "in fall prior to freeze-up (sometime in September-November), and in spring following snowmelt (sometime in April-May).

RC2.15 5,18: as hourly data does not include precipitation or snow depth perhaps reword this sentence so it is clear. AC2.15 Thank you. MC2.15 Edited to "These data include daily (1962-present) precipitation (snowfall and rainfall), temperature, wind speed and direction, and snow depth, and hourly (1995-present) temperature, wind speed and direction, and relative humidity."

RC2.16 6,7: 'snowpack' characteristics? AC2.16 As above, we mean snowpack depth, density and water equivalent MC2.16 Edited to clarify that here too.

RC2.17 Table 1: hourly meteorology data? To what set of data does the '1994-2011+ (ongoing)' refer to? AC2.17 Thanks for picking up on that MC2.17 Edited the daily and hourly data to match the relevant data, as per two comments ago. Also moved the 1994-2011+(ongoing) (corrected to 1995) to match the same line as the hourly data.

RC2.18 Figure 2: some of the light (less than 10 mm) daily runoff values are very difficult to discern. Although it is realized that the figure is for example purposes only it is important to note some of the occurrences of runoff, especially during the warm season. Suggest that the color shading start at a higher value of blue or that two colors are used, not white and/or a note about what value is considered 'measurable' on this figure. AC 2.18 Thanks for this feedback. Both you and Referee 1 thought the same, and I agree this figure can be much improved. MC 2.18 Created a new figure with improved, dual colour shading to see the small runoff events. Also created an inset of just spring snowmelt events and an inset of just summer events, to better see those events.

RC2.19 DATA REVIEW only refers to the English web links, except where noted. 1962-2011 DATA ON SOILS, RUNOFF, AND SNOW. All data was checked for outliers. Phosphate- P: Data: Watershed 1 1991 has values an order of magnitude higher than all other values. Runoff: should mention in the manuscript that daily peak flow values are also available as this information can be very useful. Runoff data: 'NA' occurs in the Runoff and Peak columns not the Flag Indicator column. The flag for the 'NA' values is 'm'. Snow Water Equivalent: snow Density units are listed as 'Mg/cm3'. Should be 'g/cm3' according to the format given in the data file. French version 'mg/cm3' should also be 'g/cm3'. Soil Moisture Content: depths are wrong for d60_d90 and d90_d120. Ok for French version. Within dataset some values for the deeper intervals are very low (for soils with clay contents greater than 20%) and must be due to sand pockets. Soil Nitrate Phosphorus: Dictionary: Depth

intervals, some errors in both English and French versions. Soil Nutrients: English Dictionary: French version comes up. Soil Nutrients Dataset: years and depth should be in same format of other soil files (eg Soil_Nitrate_Phosphorous). Watershed Management: Dictionary: System, n is missing from word 'rotation'. Watershed Management Data: under 'System' Column there is 'wheat-GM fallow'. This should be 'wheat-green manure fallow' according to the Dictionary and to the 'Management Systems Details' data set. Some cells are lacking information where it is expected; for example: Column D (Previous Crop) rows 33, 127-133; Column E rows 51, 99, 151. METEOROLOGICAL DATA. Dates of start and finish and parameters were noted for daily and hourly data. Data was only spot reviewed. No issues were noted. DIGITAL ELEVATION DATA. The description and README.txt file was read. Data was not reviewed. Description file ; 2nd last sentence; "(random toughness)"? AC2.19 Thank you for also taking the time to review the datasets, and for picking up on those errors and queries. The datasets are published separately to this data note (they have their own dois), therefore corrections to the datasets are via a separate process. We are working on instituting the following edits to the datasets: correcting Phosphate-P data, correcting the snow density units, correcting the soil moisture content depth intervals, correcting the errors in the depth intervals of the soil nutrients, correcting the link to the soil nutrients English dictionary, ensuring consistency in depth and year formats between files, correcting spelling mistakes, and ensuring data or missing data notations are in all cells. MC2.19 In this data note manuscript, we have added information in the Runoff section of the manuscript referring to the daily peak flow values that are available in the dataset. We have also described that missing data for runoff and peak runoff are shown by an NA, and that missing data is flagged with an 'm'.

Please also note the supplement to this comment:
https://www.earth-syst-sci-data-discuss.net/essd-2018-126/essd-2018-126-AC1-supplement.pdf

---

## Author Response (AR2)

**Author responses to topical editor on "Fifty years of recorded hillslope runoff on seasonally-frozen ground: The Swift Current, Saskatchewan, Canada dataset" by Anna E. Coles et al.**

14 August 2019

**Topical Editor Decision: Publish subject to technical corrections** (06 Aug 2019) by W.D. Helgason
**Comments to the Author:**
The authors have provided adequate responses to all of the the reviewer comments. This manuscript is suitable to be published after a very minor revision, which is to label Figure 2 panels as A, B, and C, and to put a note in the caption that explains the differences in time resolution. Further review will not be required.

**Comments from the Author to the Topical Editor:**
Thank you for your final review.
We have made the requested edits: we have edited Figure 2 to label the panels as A, B, and C, and we have edited Figure 2's caption to explain the difference in the time resolution.